# Identification, Phylogeny, Divergence, Structure, and Expression Analysis of A20/AN1 Zinc Finger Domain Containing *Stress-Associated Proteins* (*SAPs*) Genes in *Jatropha* *curcas* L.

**DOI:** 10.3390/genes13101766

**Published:** 2022-09-30

**Authors:** Abdul Jalal, Qurban Ali, Hakim Manghwar, Daochen Zhu

**Affiliations:** 1Biofuels Institute, School of the Environment and Safety Engineering, Jiangsu University, Zhenjiang 212013, China; 2Key Laboratory of Integrated Management of Crop Diseases and Pests, Ministry of Education, Department of Plant Pathology, College of Plant Protection, Nanjing Agricultural University, Nanjing 210095, China; 3Lushan Botanical Garden, Chinese Academy of Sciences, Jiujiang 332000, China

**Keywords:** abiotic stress, divergence, expression profile, jatropha, phylogeny, *SAP* genes

## Abstract

Jatropha is a small woody perennial biofuel-producing shrub. *Stress-associated proteins* (*SAP*s) are novel stress regulatory zinc-finger proteins and are mainly associated with tolerance against various environmental abiotic stresses in Jatropha. In the present study, the *JcSAP* gene family were analyzed comprehensively in *Jatropha curcas* and 11 *JcSAP* genes were identified. Phylogenetic analysis classified the *JcSAP* genes into four groups based on sequence similarity, similar gene structure features, conserved A20 and/or AN1 domains, and their responsive motifs. Moreover, the divergence analysis further evaluated the evolutionary aspects of the *JcSAP* genes with the predicted time of divergence from 9.1 to 40 MYA. Furthermore, a diverse range of cis-elements including light-responsive elements, hormone-responsive elements, and stress-responsive elements were detected in the promoter region of *JcSAP* genes while the *miRNA* target sites predicted the regulation of *JcSAP* genes via a candid *miRNA* mediated post-transcriptional regulatory network. In addition, the expression profiles of *JcSAP* genes in different tissues under stress treatment indicated that many *JcSAP* genes play functional developmental roles in different tissues, and exhibit significant differential expression under stress treatment. These results collectively laid a foundation for the functional diversification of *JcSAP* genes.

## 1. Introduction

Plant growth and productivity is severely affected by various biotic and abiotic stresses because of their sessile nature. These environmental stresses down- or upregulate a large pool of genes. To eliminate or reduce the harsh environmental effect, plants have evolved complex internal molecular mechanisms to modulate stress-responsive or regulatory genes [1,2]. Regulatory genes code for sensors that perceive stress signals, kinases that transmit the signals, and transcription factors that are down- or upregulated as a result of perceived stress. Thus, defence mechanisms start with the perception of stress, followed by signal transduction, synthesis of transcription factors, and finally down- or upregulation of genes that produce protective proteins and metabolites [2]. Hitherto, a large number of genes have been characterized that play an important role during different stresses and are involved at different levels of regulation, such as perception, signalling, transcription, and production of protective biomolecules [3,4,5,6,7]. The characterisation of genes belonging to the signal transduction and transcription factor categories is of great importance because of their effect on a wide range of stress-related genes. Zinc finger proteins are associated with this category and are considered the important proteins that contribute to protection against environmental stress [8,9,10,11].

Stress-associated proteins (SAP) are the newly identified stress regulatory zinc-finger proteins in plants with the A20 domain at N-terminal and AN1 domain at C-terminal, and are mainly associated with tolerance against various environmental abiotic stresses [1,12,13,14,15]. *SAP* genes in plants were first identified in *Oryza sativa*, and *OsSAP1* was declared to be largely involved in abscisic acid, healing wounds, heavy metals, salt, drought, and cold [9]. Similarly, other *OsSAP* genes have also been expressed under several environmental stresses, indicating their role in stress-response [5]. Recently, the response of *SAP* genes has been identified in several plant species and has been studied against various environmental stresses. The evidence from the previous literature has shown the presence of the *SAP* gene family in many species, like 18 *SAP* gene members in rice [5], 14 in Arabidopsis [5], 17 in barley [13], 27 in soybean [12], 37 in cotton [16], 13 in tomato [2], 12 in cucumber [17], 21 in eggplant [1], and 9 in almond [18]. However, the identification of *SAP* genes in Jatropha genome is not systematically deliberated.

Jatropha (*Jatropha curcas* L.) is a small woody perennial shrub that belongs to the subfamily Crotonoideae of Euphorbiaceae. Jatropha is a biofuel-producing non-food crop mostly grown in subtropical as well as tropical regions, with 40 to 50% of oil content in its seeds. Particularly important is the fact that Jatropha can grow on degraded soils, making it an attractive crop for biodiesel feedstock as it can be widely planted on marginal land considered inappropriate for food crops [19]. In addition, Jatropha is adaptable to grow in a wide range of agro-climatic conditions and possesses several properties like easy propagation, short gestation period, rapid growth, high oil content, low seed cost, and tolerance to salt and drought, making it suitable for biodiesel production [19,20,21]. With the increasing costs, gradual depletion of fossil energy resources, and having a strong potential for biofuel production, Jatropha is recently attracting the widespread attention of researchers. The identification of the *SAP* gene family in Jatropha might encompass a novel understanding of the flow and appearance of the mechanism of stress regulatory genes. In the current study, we identified the *SAP* gene family genome-wide in Jatropha and comprehensively analysed their phylogeny, domain, motif, gene structure, chromosome location, gene duplication, cis-acting elements, *miRNA* target sites, and the expression analysis in various tissues against stress, which laid a foundation for further studies on the biological function of *SAP* genes in Jatropha.

## 2. Materials and Methods

### 2.1. Identification, Sequence Alignment, and Phylogenetic Analysis of SAP Genes in Jatropha Curcas

To identify *SAP* genes in Jatropha, Pfam server (http://pfam.xfam.org/) (accessed on 31 August 2022) was searched for the A20 domain (PF01754) and AN1 domain (PF01428), and then the HMMER (https://www.ebi.ac.uk/Tools/hmmer/search/hmmsearch) (accessed on 31 August 2022) Hidden Markov model was used as a probe to screen all the candidate proteins. To ensure the reliability of the sequences and to remove redundant sequences, the search results of all candidate *SAP* protein sequences were further searched for the presence of A20/AN1 domains using the Pfam database (http://pfam.janelia.org/) (accessed on 31 August 2022), MOTIF search (https://www.genome.jp/tools/motif/) (accessed on 31 August 2022), NCBI conserved domain database (http://www.ncbi.nlm.nih.gov/Structure/cdd/wrpsb.cgi) (accessed on 31 August 2022), SMART database (http://smart.embl-heidelberg.de/) (accessed on 31 August 2022), and Inter ProScan program (http://www.ebi.ac.uk/Tools/pfa/ iprscan5/) (accessed on 31 August 2022). The protein sequences of the *SAP* gene family of rice and Arabidopsis (Appendix A) were retrieved from the previously published report [5]. Multiple sequence alignments of all the sequences of Jatropha, Arabidopsis, and rice were performed using Muscle. Subsequently, to create a phylogenetic tree, the alignments were imported to MEGA7 software (https://www.megasoftware.net/home) (accessed on 1 September 2022) using the neighbor-joining (NJ) method with a bootstrap option of 1000 replications. The phylogenetic tree was further visualized and edited via MEGA7 software (https://www.megasoftware.net/home) (accessed on 1 September 2022) [22,23,24,25,26].

### 2.2. JcSAP Protein Physicochemical Characteristics

To find the physicochemical properties of *JcSAP* genes, the NCBI database (https://www.ncbi.nlm.nih.gov/) (accessed on 1 September 2022) was searched to find the amino acid (bp), CDS (bp) and location on Chromosome. The ProtParam tool of ExPASy server (http://web.expasy.org/protparam) (accessed on 1 September 2022) was searched to predict the isoelectric points (pI), molecular weight (MW), grand average of hydropathicity (GRAVY), and Formula of each *SAP* gene in *Jatropha curcas*. Softberry server for plant protein location (http://www.softberry.com/berry.phtml?topic=protcomppl&group=programs&subgroup=proloc) (accessed on 1 September 2022) was employed to predict the subcellular localization each *SAP* protein in *Jatropha curcas* [22].

### 2.3. Gene Duplication Events, Homology and Synteny Analysis of JcSAP Genes

For Gene conservation, duplication events, homology, and Synteny analysis, a comparative Synteny analysis was performed by using circoletto Tool (https://www.tools.bat.infspire.org/circoletto/) (accessed on 2 September 2022) to visualize genome conservation. Protein sequences of Arabidopsis *SAP* genes were used against the identified *JcSAP* sequences and were finally exhibited by circus by running the Circoletto tool [27].

### 2.4. Conserved Domains, Motifs, and Gene Structure Organization of JcSAP Genes

The identified *JcSAP* protein sequences were subjected to NCBI CDD online software (https://www.ncbi.nlm.nih.gov/Structure/cdd/wrpsb.cgi) (accessed on 2 September 2022) for domain analysis, and the obtained results were visualized via the TBtools software (https://github.com/CJ-Chen/TBtools) (accessed on 2 September 2022). Similarly, to analyze the *JcSAP* proteins for the conserved motifs, the protein sequences of *JcSAP* were submitted to MEME suite software 5.4.1 (https://meme-suite.org/meme/tools/meme) (accessed on 2 September 2022). Consistently, to display the gene structure organization of *JcSAP* genes, the gene structure display server (http://gsds.gao-lab.org/) (accessed on 2 September 2022) was used by submitting the CDS and genomic sequences of *JcSAP* genes [22,24].

### 2.5. Divergence Analysis

For divergence analysis of *JcSAP* genes, the server *Ka*/*Ks* calculation tool (http://services.cbu.uib.no/tools/kaks) (accessed on 2 September 2022) was used and the non-synonymous substitution per nonsynonymous site (*Ka*) and synonymous substitution per synonymous site (*Ks*) was determined by inputting the DNA sequences of *JcSAP* genes using default parameters. The divergence time was calculated by the given formula [24,28].
$$Time\;of\;divergence\;\left(T\right)=\frac{Synonymous\;substitution\;rate\;\left(dS\;\mathrm{or}\;Ks\right)}{2\times Divergence\;rate\;(6.56\times 10^{-9})}\times TMY\;(10^{-6})$$

### 2.6. Protein Structure Analysis of JcSAP Genes

To identify the structural composition of 11 *JcSAP* genes, an online tool for the prediction of secondary structure SOPMA (https://npsa-prabi.ibcp.fr/cgi-bin/secpredsopma.pl) (accessed on 2 September 2022) were used. The tertiary structure of *JcSAP* proteins were visualized via uniprot (https://www.uniprot.org) (accessed on 2 September 2022) to further support the secondary structure of *JcSAP* proteins [29].

### 2.7. Cis-Elements Analysis and Predicted miRNA Target Sites

To analyse the cis-regulatory element, the upstream region of 1500 bp of each genomic sequence of the *JcSAP* genes was submitted to PlantCARE server (https://bioinformatics.psb.ugent.be/webtools/plantcare/html/) (accessed on 5 September 2022) and was searched for the presence of cis-regulatory elements. The results were then visualized using TBtools software (https://github.com/CJ-Chen/TBtools) (accessed on 6 September 2022) [24].

For the prediction *miRNA* target sites for *JcSAP* genes, the published miRNAs of *Jatropha curcas* were downloaded from the plant *miRNA* encyclopaedia (https://pmiren.com/download) (accessed on 6 September 2022) and were then subjected to the psRNATarget database (https://www.zhaolab.org/psRNATarget/analysis?function=3) (accessed on 6 September 2022) along with the CDS of *JcSAP* genes to predict the *miRNA* target sites and interaction with *JcSAP* genes. Finally, the results were visualized using Cytoscape software (Cytoscape Consortium, San Francisco, CA, USA) (http://apps.cytoscape.org/apps/stringapp) (accessed on 7 September 2022) [24,30].

### 2.8. Gene Expression Profiling of JcSAP Genes

For *JcSAP* gene expression patterns in different plant tissues under abiotic stress, the raw expression data of leaf and root tissues subjected to drought stress was retrieved from the public database of NCBI (https://www.ncbi.nlm.nih.gov/geo/query/acc.cgi?acc=GSE61109) (accessed on 11 September 2022) under the GEO accession number GSE61109. The expression profiles of all *JcSAP* genes were exhibited as reads per kilobase per million (RPKM) values and illustrated via heat map using TBtools [30,31].

## 3. Results

### 3.1. Identification and Phylogenetic Analysis of JcSAP Genes

To gain insight into the identification and evolution of *JcSAP* genes, the Hidden Markov model profiles of the A20 domain and AN1 domain were used as a probe to screen all the candidate proteins and were then used with the protein sequences of *Arabidopsis thaliana* and *Oryza sativa* to construct a phylogenetic tree (see materials and methods). Results demonstrated that *A. thaliana*, having a genome size of 135 mb and 2n chromosome number of 10, had a total of 14 *SAP* genes following both the A20 and AN1 domain. *Oryza sativa*, having a genome size of 372 mb and 2n chromosome number of 24, had a total of 18 *SAP* genes in which 12 members were following both the A20 and AN1 domain, 5 members were following only the AN1 domain, while 1 member was following only the A20 domain. *Jatropha curcas*, having a genome size of 416 mb and 2n chromosome number of 22, had a total of 11 *SAP* genes in which 8 members were following both the A20 and AN1 domain, 3 members were following only the AN1 domain, while there were no single gene members following the A20 domain only (Figure 1B, Appendix A). To further investigate the classification and the evolutionary characteristics of the *JcSAP* proteins, an unrooted phylogenetic tree was constructed based on the *SAP* protein sequences of *Arabidopsis thaliana*, *Oryza sativa*, and Jatropha (Figure 1A). All available 34 sequences, including 14 *AtSAP*, 18 *OsSAP*, and 11 *JcSAP*, were mainly clustered into four groups. Group A contained four *JcSAP* members (*JcSAP810a*, *JcSAP810b*, *JcSAP111314*, and *JcSAP12*), clustered to *AtSAP8*, *AtSAP10*, *AtSAP11*, *AtSAP13*, and *AtSAP14*. Group B consisted of three JcSAP members (*JcSAP179*, *JcSAP3*, *JcSAP46*), clustered to *AtSAP1*, *AtSA3*, *AtSAP4*, *AtSAP6*, *AtSAP7*, and *AtSAP9.* Group C consisted of three *JcSAP* members (*JcSAP2a*, *JcSAP2b*, and *JcSAP2c*), clustered to *AtSAP2*. Group D included only one *JcSAP* member (*JcSAP5*) clustered to *AtSAP5*. These results of the comparative phylogenetic relationship predicted that the SAP members clustered together or with other species may share similar biological functions against stresses.

### 3.2. Physicochemical Characteristics of JcSAP Genes

To further get into the physicochemical characteristic of the total 11 identified *SAP* genes in Jatropha in a comprehensive manner (Figure 1), the physicochemical parameters of *JsSAP* genes including gene code, location on chromosome, amino acids and CDS length, molecular weight (MW/kDa), isoelectric point (PI), GRAVY, Formula, and Predicted subcellular localization were investigated insilico and exhibited in Table 1 and Appendix A. Results demonstrated that the amino acid length *JcSAP* genes were varied from 133 (*JcSAP2a*) to 288 (*JcSAP111314*). CDS was ranged from 402 (*JcSAP2a*) to (867) (*JcSAP111314*). Molecular weight (MW/kDa) *JcSAP* genes varied from 14686.65 kDA (*JcSAP2a*) to 31998.28 (*JcSAP111314*), while the isoelectric point (PI) ranged from 8 PI (*JcSAP179*) to 9.4 PI (*JcSAP5*). Moreover, the predicted subcellular localization revealed that all JcSAP genes are cytoplasmic expect *JcSAP111314* and *JcSAP12*, which appeared to be extracellular. The presence of each *JcSAP* gene member on a specific chromosome was not predicted in silico, however the GRAVY and formula were given in Table 1. These results provide information about the basic known parameters of the *JcSAP* genes.

### 3.3. Synteny Analysis of JcSAP Genes

To further support the identification and homologous relationship among *SAP* genes, Synteny analysis of all the identified *SAP* genes of Jatropha and Arabidopsis was subjected to the circoletto tool to make a map of comparative synteny circos (see materials and methods). The Synteny map illustrated the relationship among the *SAP* genes of Jatropha and Arabidopsis species regarding their function, expression, duplication events, and evolution (Figure 2). The sequences were placed clockwise around a circle, starting at 12 o’clock, and the ideograms were placed in order to maximally untangle the ribbons; the queries and database entries were intertwined. Ribbon colors of the map diagram represent the alignment length, visualizing the sequence similarity and identity level, i.e., blue ≤ 0.25, green ≤ 0.50, orange ≤ 0.75, red > 0.75, providing an essential first glimpse at sequence relationships. The obtained results revealed that Arabidopsis *AtSAP1*, *AtSAP7*, and *AtSAP9* showed Synteny with *JcSAP179* of Jatropha. Similarly, *AtSAP2* showed Synteny with *JcSAP2a*, *JcSAP2b*, and *JcSAP2c*. *AtSAP3* showed Synteny with *JcSAP3*. *AtSAP4* and *AtSAP6* showed Synteny with *JcSAP46*. *AtSAP5* showed Synteny with *JcSAP5*. AtSAP10 showed Synteny with *JcSAP810a*, *JcSAP810b*. *AtSAP11*, *AtSAP13*, and *AtSAP14* showed Synteny with *JcSAP111314*. *AtSAP12* showed Synteny with *JcSAP12*. Moreover, based on the color intensity, the inward and outward tangling of ribbons showed conservation and duplication events, respectively, suggesting that SAP genes were conserved in Jatropha during evolution.

### 3.4. Determination of Non-Synonymous (Ka) and Synonymous (Ks) Substitution Rate

The divergence analysis was performed to gain insight into the evolutionary aspects of *JcSAP* genes by determining the non-synonymous substitution per non-synonymous site (*Ka*) and synonymous substitution per synonymous site (*Ks*) for each pair of paralogous *JcSAP* genes according to the phylogenetic tree (Appendix A) generated by *Ka/Ks* calculation server, in order to indicate the evolutionary discretion among *JcSAP* genes (Table 2). Results with a *Ka/Ks* value of <1 each pair of paralogous genes indicated the purifying selection pressure during the evolution. The divergence time for each pair of *JcSAP* genes ranged from 9.1 to 40 million years ago (MYA) (Table 3).

### 3.5. Gene Structure, Domain and Motif Analysis of JcSAP Genes

To further explore the critical fundamental function of the identified *JcSAP* genes, the *JcSAP* were further investigated for conserved domains, gene structure organization, and motifs. Eight *JcSAP* genes like *JcSAP179*, *JcSAP2a*, *JcSAP2b*, *JcSAP2c*, *JcSAP3*, *JcSAP46*, *JcSAP5*, and *JcSAP810b* followed both the A20 domain at the N terminal and AN1 domain at C terminal, except *JcSAP111314*, *JcSAP12*, and *JcSAP810a*, which followed only the AN1 domain and had no A20 domain (Figure 3A). Similarly, the motif analysis revealed that all the *JcSAP* genes followed five conserved motifs corresponding to the A20 and AN1 domains (Figure 4A). The logos of the five identified motifs are present in Figure 4B. The demonstration of the gene structure organization further revealed the coding region (CDS) distribution (Figure 3B), indicating that *JcSAP179*, *JcSAP2a*, *JcSAP5*, *JcSAP810a*, and *JcSAP810b* have one exon, 2 *JcSAP3*, *JcSAP46*, *JcSAP111314*, and *JcSAP12* have two exons, while *JcSAP2b* and *JcSAP2c* have three exons. All these results collectively further support the conserved domain, motif, and structure and thus suggested to share similar biological functions in response to environmental abiotic stresses. 

### 3.6. Protein Structure Analysis of JcSAP Genes

To delve further into the structure of the *JcSAP* genes, the secondary and tertiary structures of the *JcSAP* proteins were visualized (Table 2, Figure 5). The secondary structure of the *JcSAP* proteins explored the way these proteins fold and coil. The secondary structure of the JcSAP proteins consisted of four main elements including the α-helix (H%), β-turn (T%), extended chain (E%), and random coil (RC%). In the secondary structure of *JcSAP* proteins, the random coil (RC%) had the highest value ranging from 49.19% (*JcSAP5*) to 68.89% (*JcSAP8*/10a), followed by the α-helix (H%) ranging from 13.2% (*JcSAP12*) to 37.84% (*JcSAP5*), again followed by the extended chain (E%) ranging from 10.27% (*JcSAP5*) to 13.71% (*JcSAP12*), and β-turn (T%) ranging from 1.04% (*JcSAP11/13/14*) to 4.68% (*JcSAP2b*). Similarly, the tertiary structures of the *JcSAP* proteins visualized via uniprot (https://www.uniprot.org) (accessed on 2 September 2022) further supported the secondary structure of *JcSAP* proteins. 

### 3.7. Promoter Region Analysis and Prediction of miRNA Target Sites of JcSAP Genes

The promoter region analysis of the *JcSAP* genes resulted in the presence of a diverse range of cis-elements. The cis-elements identified during the promoter analysis of *JcSAP* genes were classified into three major groups/categories including light-responsive elements, hormone-responsive elements, and stress-responsive elements. The results revealed that there were 20 light responsive elements naming GT1-motif, chs-CMA1a, chs-CMA2a, AE-box, G-Box, Sp1, Box 4, MRE, I-box, GTGGC-motif, GATA-motif, TCT-motif, ATC-motif, TCT-motif, TCCC-motif, LAMP-element, 3-AF1 binding site, ACE, AT1-motif, GA-motif, 10 hormone responsive elements naming GARE-motif, TCA-element, TGACG-motif, CGTCA-motif, ABRE, TATC-box, TGA-element, P-box, AuxRR-core, AuxRE, and 12 Stress Responsive Elements naming TC-rich repeats, MBS, LTR, ARE, GCN4_motif, RY-element, HD-Zip 1, O2-site, WUN-motif, CAT-box, NON-box, CAT-box (Figure 6A). Moreover, the mean values of light-responsive elements ranged from 1.20 (*JcSAP810b*) to 3.50 (*JcSAP2a*), the mean values of hormone responsive elements ranged from 1.13 (*JcSAP111314*) to 4.00 (*JcSAP2b*), and the mean values of stress responsive elements ranged from 1.00 (*JcSAP2a*) to 2.33 (*JcSAP810a* and *JcSAP2c*) (Figure 6B).

Similarly, to predict the *miRNA* target sites of the *JcSAP* genes, the coding sequences of the identified *JcSAP* genes were used against the *miRNAs* of Jatropha (see material and methods). The results revealed that only three *miRNAs* (*Jcu*-*miR393*, *Jcu*-*miR1628*, and *Jcu*-*miR2111*) showed interaction with only four *JcSAP* genes (*JcSAP2a*, *JcSAP2b*, *JcSAP810a*, and *JcSAP12*) (Figure 6C). Moreover, *miRNA Jcu*-*miR393* showed interaction with two *JcSAP* genes, *JcSAP2a* and *JcSAP2b*, *miRNA Jcu*-*miR1628* showed interaction with *JcSAP12*, while *miRNA Jcu*-*miR2111* showed interaction with *JcSAP810a*. The Excel spreadsheet containing targeting sites predicted *miRNAs* ID, and the alignment with *JcSAP* gens are given in the Appendix A.

### 3.8. Gene Expression Profiling of JcSAP Genes

To investigate the possible function of *JcSAP* genes during environmental abiotic stress in Jatropha, the expression data were analysed from leaf and root tissues of Jatropha plants subjected to drought stress [32] (Figure 7A). The heat map showed the expression level of 11 *JcSAP* genes in different tissues of leaf and root during drought stress and after recovery (Figure 7B). The results further revealed that *JcSAP179*, *JcSAP3*, and *JcSAP2b* were highly expressed in leaf and in root, followed by *JcSAP111314*, *JcSAP810a*, and *JcSAP810b* as compared with the other *JcSAP* genes. Contrarily, *JcSAP2a*, *JcSAP5*, *JcSAP2c* were highly expressed in roots only. These results provide the possible involvement of *JcSAP* genes in abiotic stress tolerance in Jatropha.

## 4. Discussion

*Stress-associated proteins (SAP)* are novel stress regulatory zinc-finger proteins and are strongly associated with tolerance against various abiotic stresses [1,12,13,14,15]. *SAP* gene family has previously identified and comprehensively studied in many plant species including *Oryza sativa* [5], *Arabidopsis thaliana* [5], *Hordeum vulgare* [13], *Glycine max* [12], *Gossypium hirsutum* [16], *Solanum lycopersicum* [2], *Cucumis sativus* [17], *Solanum melongena* [1], and *Prunus dulcis* [18], etc. However, there is no systematic study of *SAP* genes in the most important biofuel-producing shrub, Jatropha. In present study, 11 *SAP* genes were identified genome-wide in Jetropha and the phylogenetic divided the *JcSAP* genes into four groups (Figure 1); Synteny analysis showed that Jatropha *SAP* genes had a high homology with the Arabidopsis *SAP* genes (Figure 2). These results are highly in consistence with the previously reported results of [1,2], suggesting very little variation in *SAP* gene family members.

To obtain further insight into the similar features and biological functions of *JcSAP* genes, a search for the conserved domain, motif, and structure was conducted (Figure 3 and Figure 4), resulting in the presence of A20 or AN1 domains and their respective motifs, thus suggesting that they share a similar biological function in response to stresses. Zinc-finger A20 or AN1 domains are highly conserved in all plant species and these results are in agreement with the previous reports [15,33]. Moreover, the domain organization revealed some domain-wise grouping, illustrating the presence of solely the AN1 domain in three *JcSAP* genes, with the others having both the A20 and AN1 domains (Figure 3). Except for Arabidopsis and tomato, similar reports were also exhibited in some other plant species like *Amborella trichopoda*, soybean, rice, and eggplant [1]. Such results may be due to the existence of a homology structure beyond the domain sequences [2]. These cases may also indicate the ancient origin of the specific AN1 domain with respect to its characteristics as compared with the A20 domain, or may also be due to the loss of one domain during evolution, as such cases occur in prologue genes (Table 2) and were also in line with the previous report on *Brassica napus* [14]. Moreover, the presence of corresponding motifs of the A20 and AN1 domain and the diversity of exon in the gene structure organization further strengthen the understanding of the evolutionary mechanisms in the *JcSAP* gene members [34,35]. In the present study, no intron has been found in *JcSAP* genes (Figure 3B), and this intron-free characteristic of *SAP* genes is usually found in other plant species like eggplant, rice, and apple [1,5,12,36]. It may be attributed to the fact that intron-free gene families can reduce posttranscriptional processing and rapidly adjust transcript expression [37].

The differential expression pattern of *JcSAP* genes in leaf and root tissues against drought stress revealed a potential role of these genes in stress response (Figure 7). Various studies have declared the role of *SAP* genes in different biotic and abiotic stresses [12,13]. Previous studies indicated that *SAP* genes play a great role in mediating abiotic stresses including cold, salt, and drought [1,2]. Our results are also in line with the previous studies on other species [12,36]. Altogether, the present study provides a baseline for understanding the molecular role of *JcSAP* genes and for further study of these genes against different abiotic stresses.

## 5. Conclusions

In conclusion, a total 11 *SAP* genes were identified during this study of Jatropha and were divided into four groups based on the phylogenetic analysis and amino acid sequence similarity; they may share similar biological functions against stresses. The physicochemical properties of *JcSAP* genes uncovered the basic gene parameters like amino acids and CDS length, molecular weight (MW/kDa), isoelectric point (PI), GRAVY, and molecular formula, and further revealed that most of the *JcSAP* genes are cytoplasmic, however, no detailed information was found regarding the chromosomal localization of *JcSAP* genes. The Synteny analysis showed that most of the *JcSAP* proteins were highly homologous to the Arabidopsis SAP proteins, indicating that *SAP* genes are conserved in Jatropha during evolution. Further domains and motifs analysis revealed that the A20 and AN1 domains are conserved in *JcSAP* genes and their similar gene structure features may be due to the duplication events during evolution. The divergence analysis further provided insight into the evolutionary aspects of *JcSAP* genes revealing the time of divergence from 9.1 to 40 MYA. The promoter region analysis of *JcSAP* genes resulted in a diverse range of cis-elements including light-responsive elements, hormone-responsive elements, and stress-responsive elements. The predicted *miRNA* target sites revealed that *JcSAP* genes may be regulated by a complicated *miRNA* mediated posttranscriptional regulatory network. In addition, the expression profiles of *JcSAP* genes in different tissues and stress treatments indicated that many *JcSAP* genes play functional developmental roles in different tissues, and exhibit significant differential expression under different stress treatments. All these results collectively suggested that *JcSAP* genes share similar biological functions in response to stresses and provide valuable clues for further investigation of *JcSAP* genes’ function and diversity.

## Figures and Tables

**Figure 1 genes-13-01766-f001:**
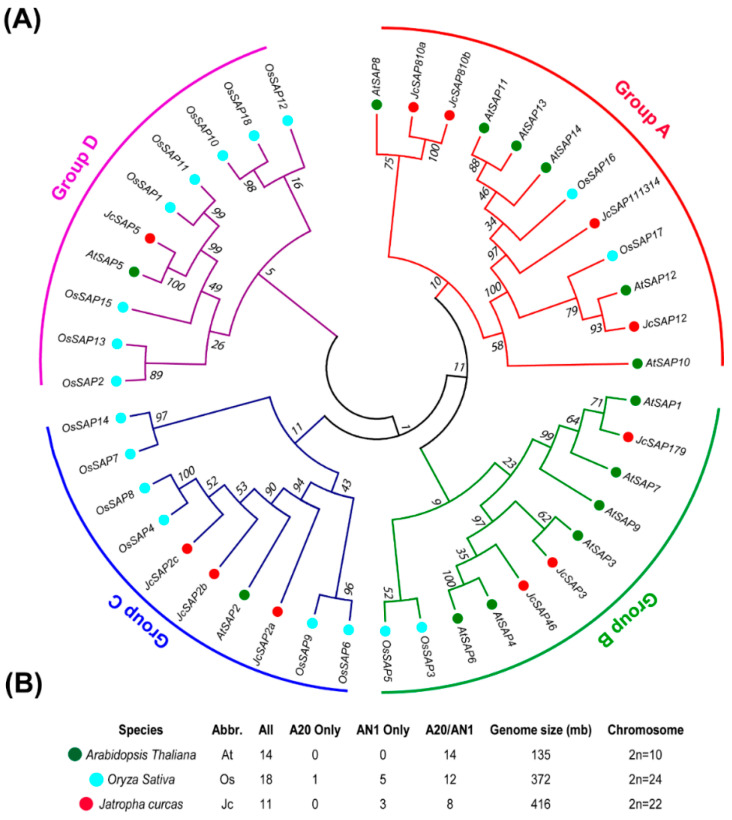
Identification and phylogenetic Analysis of *SAP* members of Jatropha against Arabidopsis and rice. (**A**) Phylogenetic tree of 14 arabidopsis, 18 rice, and 11 Jatropha *SAP* members, (**B**) Identified *SAP* members of Arabisopsis, rice, and Jatropha with followed domains, genome size, and chromosome numbers.

**Figure 2 genes-13-01766-f002:**
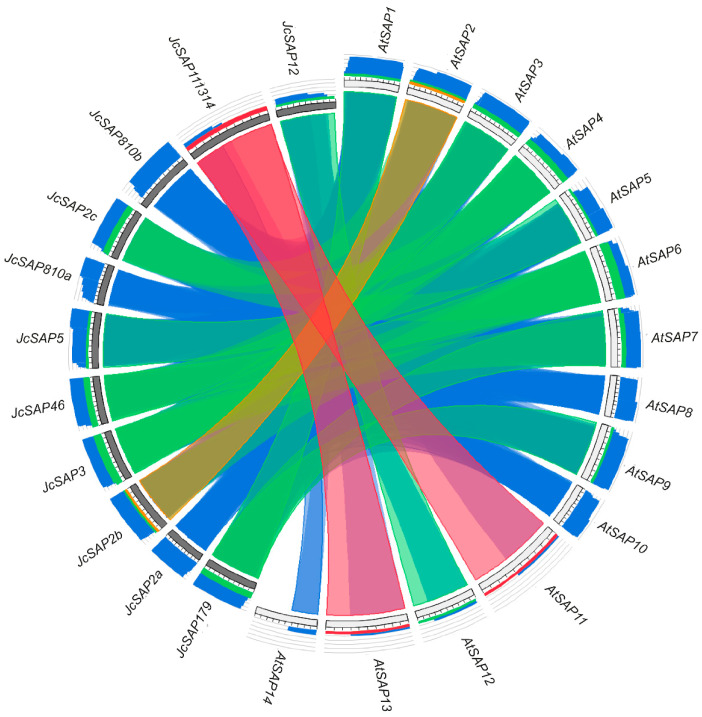
Synteny map of identified SAP genes in *Arabidopsis thaliana* and *Jatropha curcas*.

**Figure 3 genes-13-01766-f003:**
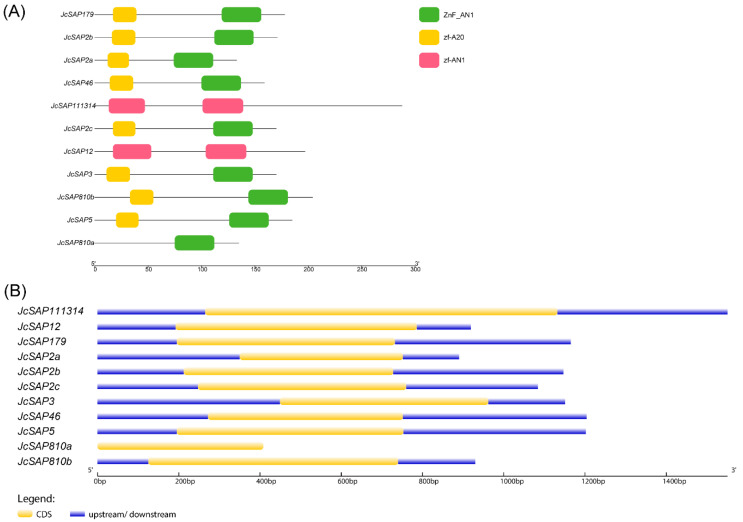
Domain (**A**) and Gene structure organization (**B**) of *JcSAP* genes.

**Figure 4 genes-13-01766-f004:**
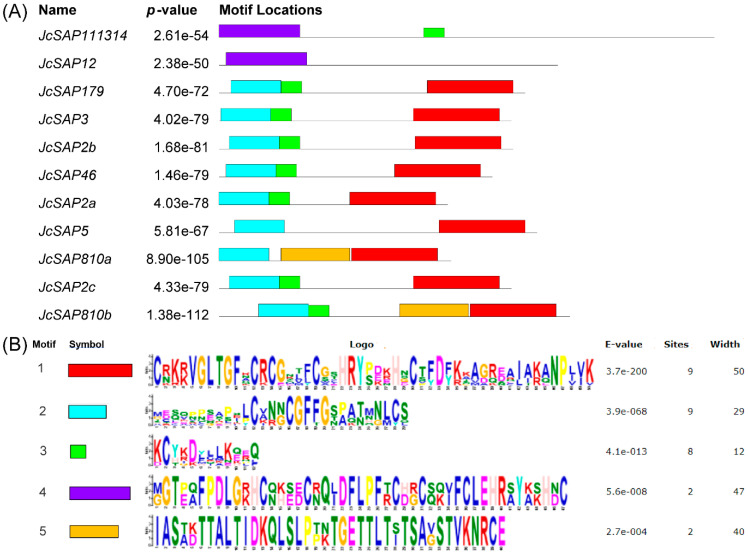
Conserved motifs of *JcSAP* genes in *Jatropha curcas*. (**A**) Identified motifs; (**B**) logos of the identified motifs.

**Figure 5 genes-13-01766-f005:**
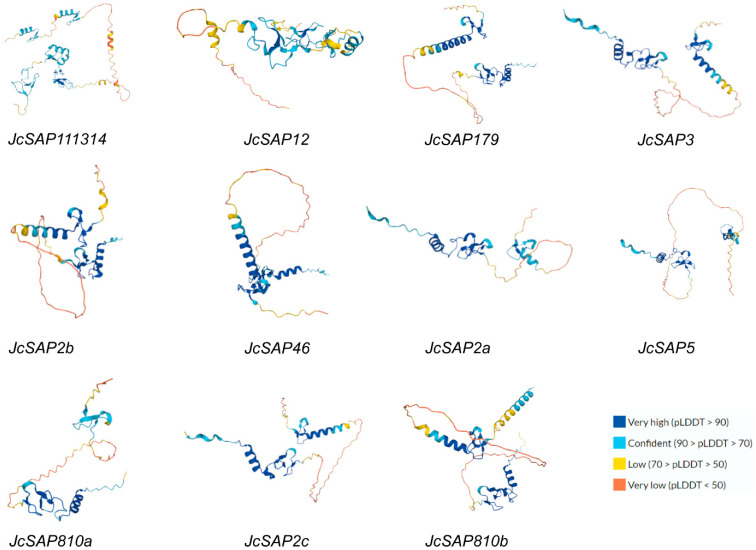
Illustration of the predicted tertiary structure of 11 *JcSAP* proteins. The protein structures all have the same domain colour schemes, revealing the degree of homology. The structures reveal a high degree of structural homology in most gene members.

**Figure 6 genes-13-01766-f006:**
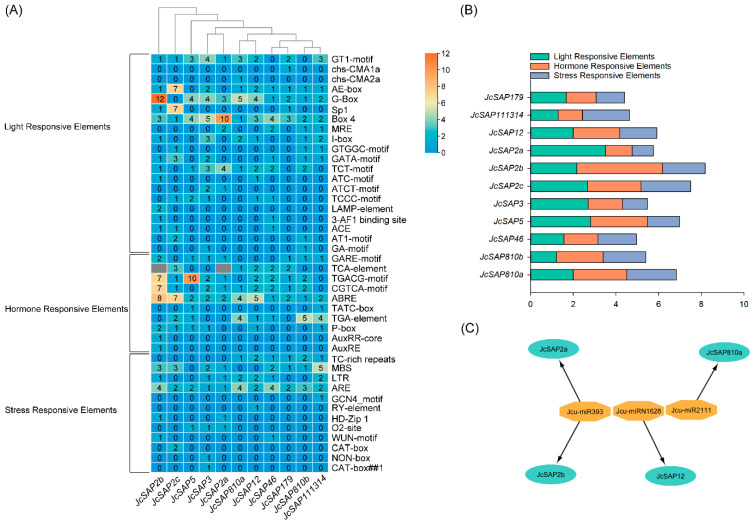
Cis regulatory elements and prediction of *miRNA* target sites of *JcSAP* genes. (**A**) The heat map of the Cis elements and their classification, present in the promoter region of *JcSAP* genes, (**B**) Graphical visualization of the groups of Cis elements in the promoter region of *JcSAP* genes, (**C**) Predicted *miRNA* target sites of *JcSAP* genes.

**Figure 7 genes-13-01766-f007:**
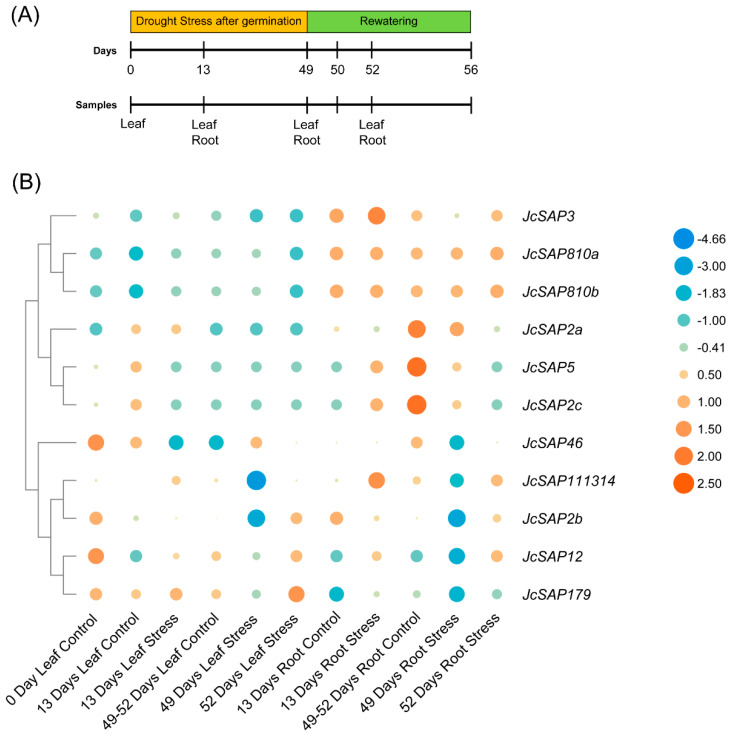
Expression levels of *JcSAP* genes in leaf and root tissues under abiotic stress. (**A**) Drought stress subjection and samples collected at different intervals during and after recovery of heat stress, (**B**) The expression levels the *JcSAP* genes at different intervals of leaf and root with their respective controls.

**Table 1 genes-13-01766-t001:** Physicochemical characteristics of *JcSAP* genes.

Gene Name	Gene ID (Uniprot)	Chr	AA	CDS	MW/kDa	PI	GRAVY	Formula	Predicted Subcellular Localization
*JcSAP179*	A0A067JZZ9_JATCU	Un	178	537	18834.23	8	−0.453	C_809_H_1279_N_231_O_261_S_13_	Cytoplasmic
*JcSAP2a*	A0A067KTX6_JATCU	Un	133	402	14686.65	9.1	−0.612	C_630_H_990_N_188_O_194_S_12_	Cytoplasmic
*JcSAP2b*	A0A067JE11_JATCU	Un	171	516	18507.08	8.5	−0.442	C_785_H_1266_N_236_O_249_S_16_	Cytoplasmic
*JcSAP2c*	A0A067L4U9_JATCU	Un	170	513	18087.44	8.8	−0.304	C_761_H_1233_N_229_O_251_S_15_	Cytoplasmic
*JcSAP3*	A0A067KRF9_JATCU	Un	170	513	18661.18	8.8	−0.492	C_793_H_1276_N_236_O_254_S_15_	Cytoplasmic
*JcSAP46*	A0A067L6B6_JATCU	Un	159	480	17559.01	8.6	−0.521	C_754_H_1197_N_225_O_231_S_14_	Cytoplasmic
*JcSAP5*	A0A067L4Q5_JATCU	Un	185	558	20311.78	9.4	−0.665	C_849_H_1395_N_271_O_283_S_12_	Cytoplasmic
*JcSAP810a*	A0A067JER8_JATCU	Un	135	408	14879.04	9.3	−0.49	C_643_H_1038_N_186_O_199_S_10_	Cytoplasmic
*JcSAP810b*	A0A067JQU7_JATCU	Un	204	615	22711.95	9.3	−0.775	C_972_H_1602_N_282_O_314_S_14_	Cytoplasmic
*JcSAP111314*	A0A067K2L3_JATCU	Un	288	867	31998.28	8.6	−0.665	C_1367_H_2182_N_416_O_426_S_23_	Extracellular
*JcSAP12*	A0A067LAI2_JATCU	Un	197	594	21773.79	9	−0.639	C_924_H_1487_N_285_O_288_S_18_	Extracellular

**Table 2 genes-13-01766-t002:** Secondary Structure of 11 *JcSAP* proteins.

	H (%)	T (%)	E (%)	RC (%)
JcSAP11/13/14	22.22	1.04	11.11	65.62
JcSAP12	13.2	2.54	13.71	70.56
JcSAP1/7/9	21.35	2.81	11.8	64.04
JcSAP3	27.65	3.53	11.76	57.06
JcSAP2b	25.15	4.68	10.53	59.65
JcSAP4/6	30.19	4.4	12.58	52.83
JcSAP2a	26.32	4.51	13.53	55.64
JcSAP5	37.84	2.7	10.27	49.19
JcSAP8/10a	14.07	3.7	13.33	68.89
JcSAP2c	28.82	4.12	11.76	55.29
JcSAP8/10b	21.08	2.94	13.24	62.75

**Table 3 genes-13-01766-t003:** Non-synonymous (*Ka*) and synonymous (*Ks*) substitution rate and divergence time of *JcSAP* genes.

Paralogous Genes		*Ka*	*Ks*	*Ka/Ks*	T (MYA)
*JcSAP111314*	*JcSAP12*	0.2614	0.5275	0.495545	40.2
*JcSAP810b*	*JcSAP810a*	0.04609	0.11945	0.385851	9.1
*JcSAP2b*	*JcSAP2c*	0.12225	0.33925	0.360354	25.8
*JcSAP4/6*	*JcSAP3*	0.1221	0.4072	0.299853	31.0
*JcSAP5*	*JcSAP179*	0.26015	0.46745	0.55653	35.6

## Data Availability

The data supporting the results are already mentioned in the main text and in Appendix A.

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
