# Peer review of "Identification, Phylogeny, Divergence, Structure, and Expression Analysis of A20/AN1 Zinc Finger Domain Containing Stress-Associated Proteins (SAPs) Genes in Jatropha curcas L."

_genes, 2022, doi:10.3390/genes13101766_

Round 1

Reviewer 1 Report

The authors of this article identified stress-associated proteins (SAP) from Jatropha curcas which are stress-regulatory zinc-finger proteins and are found to be associated with various abiotic stresses. They further performed sequence alignment, phylogenetic analysis, and studied the physiochemical characteristics of these proteins. They also performed homology and synteny analysis as well as studied the conserved domains, gene structure, and analyzed the motifs and protein structure. Finally they analyzed cis-elements, predicted miRNA target sites and studied the gene expression profiling of the jcSAP genes. This study forms the basis of further molecular characterization of these genes in order to better understand the plants response to abiotic stress. The methods, results and discussion are decribed well. However, the manuscript needs extensive grammatical correction.

Please be consistent with writing scientific names and make sure they are italicized.

Sometimes JsSAP is italicized while many times it's not. 

Line 143: "Then, the tertiary structure of was....." correct the sentence

Line 144: visualized is repeated twice.

Line 164: correct the sentence

In line 219, the authors mention "Synteny map illustrates an incredible relationship among SAP....." There is a relationship between the Arabidopsis and Jatropha SAP genes but I am not sure of incredible in this context.

Line 312: "Similarly, the to predict...." correct the sentence.

There are numerous corrections throughout the manuscript, especially in the introduction, discussion, and conclusion sections.

Author Response

Please be consistent with writing scientific names and make sure they are italicized. Sometimes JsSAP is italicized while many times it's not. 

Line 143: "Then, the tertiary structure of was....." correct the sentence

Line 144: visualized is repeated twice.

Line 164: correct the sentence

In line 219, the authors mention "Synteny map illustrates an incredible relationship among SAP....." There is a relationship between the Arabidopsis and Jatropha SAP genes but I am not sure of incredible in this context.

Line 312: "Similarly, the to predict...." correct the sentence.

Response; All the highlighted sentences are revised carefully and removed the confusions.

There are numerous corrections throughout the manuscript, especially in the introduction, discussion, and conclusion sections.

Response; All the sections of the manuscript are doubled checked and the sentences are revised.

Reviewer 2 Report

This article focused on the Identification, Phylogeny, Divergence, Structure and Expression Analysis of A20/AN1 Zinc Finger domain containing Stress-Associated Proteins (SAPs) genes in Jatropha curcas L and identified 11 SAP genes in Jatropha. This study will help to facilitate stress tolerance studies in the future. Some shortcomings should be resolved before the recommendation of this article to be published.

General comments

The study is well designed and presented, but some literature is not cited. Authors are advised to cite the relevant literature where missing.

Abstract

Line 15, 16, and 24, the name of genes should be italicized.

Line 20, the motif should be changed to motifs.

Introduction

The introduction part is presented well; however, some revision is advised.

Line 52, 53, the sentence should be revised.

Line 56 to 60, recheck and revise the sentence.

Line 61, 62, family word should be added at the end of sentence.

Line 63, reference should be added.

Materials and Methods

Methodology is well presented and explained but lacks citation.

Each section should be provided with a related citation.

Line 83, the date of access to the database should be added.

Line 85, then should be removed and the start of the sentence.

Line 86, 93 SAP should be italicized.

Line 95 to 100, reference should be added.

Section 2.2, 2.3, 2.4, 2.5, 2.6, 2.7, missing the relevant citation. Reference should be provided for each section.

Results

This section is presented well, but the gene names should be italicized throughout the results section.

Line 210, the result should be changed to results.

Line 219 Tool should be changed to the tool.

Line 270 stresses should be replaced with environmental abiotic stresses.

Line 286 sentence should be revised.

Line 307, 308 the sentence should be checked for proper revision.

Discussion

The authors discussed their results well and correlated their results with the previous literature; however, the last paragraph should be removed, and the discussion for each result should be revised with justification.

Conclusion

This part is well elaborated.  

Author Response

Authors’ response to Editor’s comments;

All the comments were carefully incorporated in track changes within the main text of paper, and also respond to each comment regarding proofread in the main text of the manuscript. The similarity of the paper was also revised carefully. However, the separate responses are presented in the table below for easiness;

S/No.

Reviewers’ Comments

Authors Response

Reviewer comments

Response

1

This article focused on the Identification, Phylogeny, Divergence, Structure and Expression Analysis of A20/AN1 Zinc Finger domain containing Stress-Associated Proteins (SAPs) genes in Jatropha curcas L and identified 11 SAP genes in Jatropha. This study will help to facilitate stress tolerance studies in the future. Some shortcomings should be resolved before the recommendation of this article to be published.

The authors are grateful to the reviewer for appreciation of work and for highlighting shortcomings. All the comments are carefully incorporated in the manuscript.

2

General comments

The study is well designed and presented, but some literature is not cited. Authors are advised to cite the relevant literature where missing.

The authors cited literatures where missing.

3

Abstract

Line 15, 16, and 24, the name of genes should be italicized.

Line 20, the motif should be changed to motifs.

The mentioned lines are revised carefully.

4

Introduction

The introduction part is presented well; however, some revision is advised.

Line 52, 53, the sentence should be revised.

Line 56 to 60, recheck and revise the sentence.

Line 61, 62, family word should be added at the end of sentence.

Line 63, reference should be added.

All the highlighted sentences are revised carefully and removed the confusions.

5

Materials and Methods

Methodology is well presented and explained but lacks citation.

Each section should be provided with a related citation.

Line 83, the date of access to the database should be added.

Line 85, then should be removed and the start of the sentence.

Line 86, 93 SAP should be italicized.

Line 95 to 100, reference should be added.

Section 2.2, 2.3, 2.4, 2.5, 2.6, 2.7, missing the relevant citation. Reference should be provided for each section.

The relevant citations are incorporated to each section, accordingly.

6

Results

This section is presented well, but the gene names should be italicized throughout the results section.

Line 210, the result should be changed to results.

Line 219 Tool should be changed to the tool.

Line 270 stresses should be replaced with environmental abiotic stresses.

Line 286 sentence should be revised.

Line 307, 308 the sentence should be checked for proper revision.

All the sentences are revised carefully and removed the confusions.

7

Discussion

The authors discussed their results well and correlated their results with the previous literature; however, the last paragraph should be removed, and the discussion for each result should be revised with justification.

The discussion section is carefully revised; the mistakes are removed. Justification for each part of the result is provided along with proper references.

8

Conclusion

This part is well elaborated.

Authors are grateful.
